# Change in Physical Activity, Sleep Quality, and Psychosocial Variables during COVID-19 Lockdown: Evidence from the Lothian Birth Cohort 1936

**DOI:** 10.3390/ijerph18010210

**Published:** 2020-12-30

**Authors:** Judith A. Okely, Janie Corley, Miles Welstead, Adele M. Taylor, Danielle Page, Barbora Skarabela, Paul Redmond, Simon R. Cox, Tom C. Russ

**Affiliations:** 1Lothian Birth Cohort Studies, Department of Psychology, University of Edinburgh, Edinburgh EH8 9JZ, UK; janie.corley@ed.ac.uk (J.C.); miles.welstead@ed.ac.uk (M.W.); adele.taylor@ed.ac.uk (A.M.T.); danielle.page@ed.ac.uk (D.P.); b.skarabela@ed.ac.uk (B.S.); paul.redmond@ed.ac.uk (P.R.); simon.cox@ed.ac.uk (S.R.C.); t.c.russ@ed.ac.uk (T.C.R.); 2Alzheimer Scotland Dementia Research Centre, University of Edinburgh, Edinburgh EH8 9JZ, UK

**Keywords:** coronavirus, home quarantine, social support, physical activity

## Abstract

(1) Objectives: The COVID-19 pandemic has disproportionately affected the lives of older people. In this study, we examine changes in physical activity, sleep quality, and psychosocial variables among older people during COVID-19 lockdown. We build on cross-sectional studies on this topic by assessing change longitudinally. We also examined whether participant characteristics including demographic, cognitive, personality, and health variables were related to more positive or negative changes during lockdown. (2) Methods: 137 older participants (mean age 84 years) from the Lothian Birth Cohort 1936 study were included in the analysis. They completed the same questionnaires assessing physical activity, sleep quality, mental wellbeing, social support, loneliness, neighbourhood cohesion, and memory problems before (mostly 2 years earlier) and again during national lockdown. (3) Results: On average, levels of physical activity were reduced (those doing minimal physical activity increased from 10% to 19%) and perceived social support increased during lockdown (effect size *^d^_rm_* = 0.178). More positive change in the psychosocial and behavioural outcome variables during lockdown was associated with personality traits (greater intellect, emotional stability, and extraversion) and having a higher general cognitive ability. Participants with a history of cardiovascular disease, more symptoms of anxiety, or who lived alone were more likely to experience negative changes in the outcome variables during lockdown. (4) Discussion: These results provide further insight into the experiences of older people during the COVID-19 pandemic and could help to identify those at greatest risk of negative psychosocial or behavioural changes during this time.

## 1. Introduction

The sudden lockdown measures imposed in response to the COVID-19 pandemic (including social distancing, shielding, self-isolation, and travel restrictions) have affected many aspects of people’s lives. Cross-sectional studies of diverse populations have documented the many consequences of lockdowns across the globe. These include reduced levels of physical activity [1,2], poorer diet [3], and an increased prevalence of sleep problems [4]. Various psychosocial consequences of lockdown have also been reported, including higher rates of loneliness [5,6], less social cohesion (particularly in deprived communities) [7], more mental health difficulties, and a decrease in psychological wellbeing [8,9,10]. Potentially positive consequences of lockdown have also been described; for instance, studies of adults in Egypt, India, and the US report a moderate increase in perceived social support during lockdown [11,12,13].

Older people, specifically those over the age of 75, carry the highest risk of mortality from COVID-19 [14]. Stricter home quarantine measures and attempts to shield those at greatest risk of severe illness during lockdown mean that older people are at a greater risk of social isolation, especially those who may already have limited contact with the outside world. Accordingly, this group could be particularly affected by the COVID-19 pandemic, potentially experiencing lockdown and its consequences on mental and physical health for a longer time and to a greater degree than others [15,16]. On the other hand, there is emerging evidence from studies of European populations that compared with young and middle-aged adults, older people may be more resilient to some of the negative effects of lockdown: potentially experiencing fewer sleep problems [4] and mental health consequences [17] than their younger counterparts. However, the number of studies focusing on the experiences of older people is still limited and very few examine the broad range of potential psychosocial and behavioural consequences of lockdown for this population. A comprehensive assessment of these changes is important as they could have long lasting consequences for older people. Physical inactivity, poor sleep quality, social isolation, and low psychological wellbeing are variously associated with adverse outcomes in older age including cognitive decline, disease, and mortality risk [18,19,20,21,22,23]. In addition, with only a few exceptions [24,25,26,27,28] studies into the consequences of lockdown measures have been cross-sectional, relying on participants’ recollection of their prior states to infer how lockdown measures have affected change; objective evidence of changes in psychosocial and behavioural variables is currently limited.

As further local and national lockdowns are implemented, information regarding the experiences of older people could inform public health strategies to protect those who are most vulnerable during this time. This information will also be of value in the case of any future pandemics. The Lothian Birth Cohort 1936 (LBC1936) provides a rare opportunity to examine changes in a range of psychosocial and behavioural variables longitudinally by comparing participant responses to the same questionnaires, which were completed before and during lockdown. Participant responses were mostly collected during the most recent wave of the LBC1936 study (2017–2019) and during two weeks of lockdown as part of the LBC1936 COVID-19 survey. At the time of lockdown in Scotland, LBC1936 participants were in their mid-eighties, an age at which (for most people) social contact is important for quality of life, happiness, mental stimulation, and for many who rely on others for caring duties [29]. Using these data, we were able to examine a range of psychosocial and behavioural variables that might be affected by the experience of national lockdown. With reference to variables identified in other studies on this topic, we specifically examine changes in physical activity, sleep quality, psychological wellbeing, loneliness, social support, and neighbourhood cohesion. We also tested for changes in subjective memory problems as this experience is common in older age [30] and closely linked with the other outcome variables in the study including loneliness [31] and psychological wellbeing [32]. We also test for correlations to explore whether these variables change in tandem. The COVID-19 pandemic is likely to affect individuals differently; some people may cope well with the changes, whilst others may not. LBC1936 participants are richly-phenotyped individuals having completed five waves of assessments spanning 12 years. Using these data, we were able to examine whether participant characteristic including sex, living situation, occupational social class, childhood IQ, cognitive ability at age 82, years of education, history of depression or anxiety, and medical history were related to degree of change in psychosocial or behavioural variables during lockdown.

## 2. Materials and Methods

### 2.1. Participants

Participants were from the Lothian Birth Cohort 1936 (LBC1936) study, a community-dwelling sample of 1091 individuals, being studied in later life to ascertain the determinants of cognitive and brain ageing. All were born in 1936 and most are surviving participants of the Scottish Mental Survey of 1947 [33]. In the present study we link data collected as part of the LBC1936 study (T1) with participant responses to an online survey completed during national lockdown (T2; 27 May–8 June 2020).

### 2.2. LBC1936 Study Assessments

At study entry at age 70 (2004–2007, *n* = 1091) and subsequently every three years at ages 73 (2007–2010, *n* = 866), 76 (2011–2013, *n* = 697), 79 (2014–2017, *n* = 550), and 82 (2017–2019, *n* = 431), participants were invited to undergo repeated assessments of cognitive, medical, genetic, brain imaging (from age 73), lifestyle, and psychosocial factors. Full details on the background, recruitment, and data collection procedures are available [34,35,36]. Ethical permission was obtained from the Multi-Centre Research Ethics Committee for Scotland (Wave 1: MREC/01/0/56), the Lothian Research Ethics Committee (Wave 1: LREC/2003/2/29), and the Scotland A Research Ethics Committee (Waves 2, 3, 4, and 5: 07/MRE00/58). Written consent was obtained from participants at each wave.

### 2.3. LBC1936 COVID-19 Questionnaire

On the 27th of May 2020, 34 days following national lockdown, letters were posted to all eligible LBC1936 participants (*n* = 454) inviting them to complete the LBC1936 COVID-19 online questionnaire (created using the Qualtrics platform [37]). Participants were considered eligible if they were currently registered with the study (note that this also includes participants who did not participate in the most recent wave of the LBC1936 study at mean age 82). The LBC1936 COVID-19 online questionnaire was designed to assess participants’ knowledge and feelings about the COVID-19 pandemic and the impact of lockdown measures on their lifestyle and health. It consisted of 145 questions including questionnaires administered at previous waves of the LBC1936 study. These repeat questionnaires assessed participants’ experience of memory problems, loneliness, psychological wellbeing, social support, neighbourhood cohesion, physical activity, and sleep quality during lockdown. Participants who were unable to complete the questionnaire themselves could request that a welfare power of attorney, guardian, or nearest relative complete the survey on their behalf. The questionnaire took approximately 30 min to complete.

## 3. Measures

In the current study, we used self-reported measures of physical activity, sleep quality, and psychosocial variables, these were assessed as part of the LBC1936 COVID-19 survey (mean age 84) and previously as part of the LBC1936 study; all measures were taken from Wave 5 of the study (mean age 82) with the exception of sleep quality, which was most recently assessed at Wave 4 (mean age 79). These measures are as follows:

### 3.1. Physical Activity and Sleep Quality

Physical activity was derived from responses to a single questionnaire item, which asked: “What level of physical activity do you mainly do?”. Participants responded on a six-point scale ranging from “moving only in connection with necessary (household) chores” to “keep fit/heavy exercise or competitive sport several times per week”. Responses were scored from 1 to 6 with a higher score indicating more physical activity.

Sleep quality was also assessed with a single item adapted from the Pittsburgh Sleep Quality Index [38]: “During the last month how would you rate your sleep quality overall?”. Participants chose one of four options ranging from “very bad” to “very good”. Responses were scored on a scale of 1–4 with higher scores indicating better sleep quality. Note that sleep quality was most recently assessed at Wave 4 at mean age 79. Thus, the follow-up period for sleep quality is longer than for the other outcome variables.

### 3.2. Psychosocial Variables

Participants responded yes or no to five questions about their memory: “Do you currently have any problems with your memory?”; “If yes, are these problems interfering with your normal life?” (participants reporting no problems with their memory were automatically coded as no for this question); “Do you forget where you have left things more than you used to?”; “Do you forget the names of close friends or relatives?”; and “Have you ever been in your own neighbourhood and forgotten your way?”. Responses were coded 1 = yes, 0 = no and summed to create a score ranging from 0 to 5 with higher scores indicating a more severe subjective memory problem.

Participants completed the Warwick-Edinburgh Mental Wellbeing Scale (WEMWBS) [39]. The scale consists of 14 positively worded statements. Examples include “I’ve been feeling optimistic about the future” and “I’ve been dealing with problems well”. For each statement, respondents were asked to indicate which of five options, ranging from “none of the time” (scored 1) to “all of the time” (scored 5), best described their experience over the last two weeks. The overall score was calculated by summing the scores for each item. A higher score indicates a higher level of mental wellbeing.

Loneliness was measured with a single item: “How much of the time during the past week have you felt lonely?”. The four response options ranged between “none or almost none of the time” to “all or almost all of the time”. Responses were scored on a scale of 1–4 with higher scores indicating higher levels of loneliness.

Social support was assessed with the 7-item Perceived Social Support Scale previously used in the Health Survey for England [40]. Examples include “There are people I know amongst my family or friends who do things to make me happy” and “There are people I know amongst my family and friends who can be relied upon no matter what happens”. The three responses options were “not true” (coded as 0); “partly true” (coded as 1); and “certainly true” (coded as 2). Responses were summed to create a social support score ranging from 0 to 14 with higher scores indicating greater social support.

We used an 8-item neighbourhood cohesion questionnaire, which was adapted from the Neighbourhood Cohesion Scale [41]. Examples questions include “I feel like I belong to this neighbourhood” and “I borrow things and exchange favours with my neighbours”. Response options were on a five point-Likert scale ranging from “strongly agree” (scored 4) to “strongly disagree” (scored 0). We summed responses to create a score representing neighbourhood cohesion with possible scores ranging from 0 to 32 and higher scores indicating greater cohesion.

### 3.3. Covariates

We chose sex, age 11 IQ, years of education, occupational social class, personality traits, living situation (living alone or not), symptoms of depression and anxiety, history of diabetes or cardiovascular disease (CVD), and older age fluid cognitive ability as potential covariates of change in the psychosocial and behavioural variables. Age 11 IQ was derived from scores obtained from SMS1947 records and converted from Moray House Test scores to a standard IQ-type score (where mean = 100, SD = 15). Years of education and occupational social class were both recorded at Wave 1 of the study at mean age 70. Participants reported years of formal full-time education and their highest occupation before retirement. Occupations were then grouped into six occupational social class categories ranging from professional (coded as 1) to unskilled (coded as 5) following the Classifications of Occupations system [42]. For the main analysis, skilled manual and partly skilled manual categories were grouped together, as there were few participants in these categories (no participants were in the unskilled manual group). Most of the remaining covariates were assessed at Wave 5 of the study at mean age 82. At that wave, depression and anxiety symptoms were assessed with the Hospital Anxiety and Depression Scale (HADS) anxiety and depression subtests [43]. Participants self-reported whether they had ever been diagnosed with diabetes or cardiovascular disease (CVD). Personality traits were measured with the 50-item International Personality Item Pool (IPIP) [44], which has 10 items for each of the five factor model personality traits: emotional stability, extraversion, agreeableness, conscientiousness, and intellect. We derived a measure of fluid cognitive ability “gf” using test scores on six non-verbal subtests from the Wechsler Adult Intelligence Scale-III U.K. (WAIS-III) [45]: Matrix Reasoning, Letter Number Sequencing, Block Design, Symbol Search, Digit Symbol, and Digit Span Backward. Factor scores representing gf were estimated using confirmatory factor analysis for all participants included in the analytical sample (with complete data on all other variables in the analysis; *n* = 137) using the full information maximum likelihood (FIML) method. Finally, we used living situation (living alone or cohabiting) as reported by participants on the LBC1936 COVID-19 online survey as the most current source of this data. The timeline in Figure 1 shows when each outcome and covariate variable was assessed.

### 3.4. Analytical Sample

Participants with complete data on the outcome variables and the covariate variables (excluding the WAIS-III tests) were included in the analytical sample; *n* = 137.

### 3.5. Analysis

We firstly describe the characteristics of the participants included in the sample and report their responses on the outcome variables before (T1) and during (T2) lockdown. T1 and T2 responses were compared using paired sample *t*-tests or Wilcoxon signed rank tests (for single-item ordinal responses and variables with skewed distributions). We tested for correlations between the outcome variables within each measurement occasion (T1 and T2) using Spearman’s rho. For descriptive purposes, we calculated a raw difference score (score at T2–score at T1) for each of the outcome variables and the correlation between them using Spearman’s rho.

For the main analysis, which is illustrated in Figure 2, we estimated change between T1 and T2 for each of the outcome variables and tested whether change was associated with the covariate variables. For mental wellbeing, neighbourhood cohesion, social support, subjective memory problems, and physical activity this was achieved using a latent change score modelling approach as described by McArdle and Hamagami [46]. Using this method, latent change between two time points (T2 change) can be estimated as part of a structural equation model. In each model, we additionally regressed T2 change on the score at T1 so that the change variable was independent of the participants’ status at T1. We tested for associations with the covariate variables by regressing T2 change on each of the covariate variables in turn. The variables listed above were treated as continuous variables in the analysis. The remaining variables, sleep quality and loneliness, were assessed using single items with few response categories and were therefore not suited to latent change score analysis (it is possible to estimate a latent change scores using ordinal data; however, such models require more parameters to be estimated and are prone to convergence issues [47], as was the case here). Instead, we used responses at T1 and T2 to calculate an ordinal change variable with three categories indicating whether the participant experienced an increase, no change, or a decrease in the outcome variable. We tested for associations using ordinal regression and including each covariate in turn while additionally controlling for status at T1. The proportional odds assumption that the odds ratio (OR) is equal at each threshold of the change variable was tested with the test of parallel lines.

Following univariate analysis (testing for associations with each covariate in turn) final models, including all covariates showing significant univariate associations with the outcome (*p* < 0.05, uncorrected), were run for each outcome variable. Our analytical approach is summarised in Figure 2.

Because research into the consequences of lockdown is at an early stage, the present study was treated as exploratory [48]. We report and discuss results from unadjusted tests (with *p* < 0.05 considered statistically significant); our results therefore provide preliminary information regarding the behavioural and psychosocial consequences of lockdown. For reference, we additionally report *p*-values corrected for multiple testing. Groups of *p*-values considered to be from the same family of tests, i.e., *p*-values for covariate variables predicting the same outcome, were corrected using Hochberg’s False Discovery Rate (FDR) correction [49].

Latent change score analysis was conducted using Mplus Version 8.4 [50] and robust maximum likelihood (MLR) estimation with standard errors that are robust to non-normality. Ordinal regression analysis and data preparation, plotting, and calculation of descriptive statistics was conducted in the R software environment, version 3.6.1 [51].

## 4. Results

Appendix A compare participants that were included and excluded from the analytical sample on the covariate variables. Excluded participants are individuals who took part in Wave 5 of the LBC1936 study but were not included in the analytical sample (either because they did not respond to the survey or because they had missing data). Participants included in the analytical sample, on average, had a higher age 11 IQ (*p* = 0.001) and scored higher on the tests of fluid cognitive ability at age 82 than participants excluded from the sample (*p* range = 0.01 and <0.001). Included participants also tended to have more years of education (*p* = 0.002) and a higher occupational social class (*p* < 0.001) and score higher on the personality traits intellect (*p* = 0.020) and emotional stability (*p* = 0.031). Appendix A compares participants included and excluded from the analytical sample on the outcome variables assessed at baseline (mean age 82 or 79; T1). Participants included in the analytical sample, on average, reported less severe memory problems (*p* = 0.046); however, they did not differ from participants excluded from the analytical sample on any of the other outcome variables.

### 4.1. Descriptive Results

Table 1 shows participants’ responses on the outcome variables at baseline (mean age 82 or 79; T1) and during lockdown (mean age 84; T2). On average, participants did less physical activity (*p* = 0.012) and reported slightly higher levels of social support during lockdown than a few years beforehand (*p* = 0.032). To examine change in physical activity further, we additionally performed McNemar’s test (with continuity correction) comparing the number of participants reporting minimal physical activity (only household chores) vs. any higher category of physical activity (outdoor activities 1–2x per week or more) before and during lockdown. The higher number of participants reporting minimal physical activity during lockdown was statistically significant (*p* = 0.038).

Figure 3 shows the distribution of participant responses at T1 and T2; the boxplots also illustrate the variance of within participant change between T1 and T2. 

### 4.2. Correlations between Outcomes

Appendix A show correlations between the outcome variables at baseline (T1) and during lockdown (T2). Table 2 shows correlations between changes (between baseline and lockdown) in the outcome variables. A decrease in wellbeing was correlated with an increase in loneliness (*r* = −0.220) and memory problems (*r* = −0.223). A decrease in social support was correlated with an increase in memory problems (*r* = −0.169) and a decrease in wellbeing (*r* = 0.174).

### 4.3. Predictors of Change

Results from univariate latent change score and ordinal regression models (entering each covariate in turn) predicting change in wellbeing, social support, physical activity, neighbourhood cohesion, memory problems, loneliness, and sleep quality are shown in Appendix A. Covariates that were significantly related to changes in these six outcome variables were entered simultaneously into final models. These results are displayed in Table 3.

In these final models, more positive change in mental wellbeing was associated with higher emotional stability (*β* = 0.221, *p* = 0.001) and more negative change in mental wellbeing was associated with history of CVD (*β* = −0.376, *p* = 0.004). More positive change in neighbourhood cohesion was associated with higher intellect (*β* = 0.147, *p* = 0.021) and more negative change was associated with history of CVD (*β* = −0.310, *p* = 0.025). More positive change in social support was associated with higher intellect (*β* = 0.118, *p* = 0.044) and being in a manual (skilled or unskilled) occupational class relative to a professional occupational class (*β* = 0.575, *p* = 0.003). More positive change in physical activity was associated with higher extraversion (*β* = 0.162, *p* = 0.012) and a higher age 11 IQ (*β* = 0.143, *p* = 0.026). More positive change in memory problems (i.e., an increase in memory problems) was marginally associated with a lower score on the HADS depression subscale (*β* = −0.159, *p* = 0.047) but not with any of the other covariate variables. In the final ordinal regression model predicting change in loneliness, a more positive change in loneliness (an increase in loneliness) was associated with higher scores on the HADS anxiety subscale (OR = 1.222, *p* = 0.010); living with others was associated with a more negative change (decrease) in loneliness (OR = 0.149, *p* < 0.001).

### 4.4. Subsidiary Analysis

In the model predicting change in loneliness, the proportional odds assumption was not met for HADS anxiety score or living situation, indicating that the relationship between these variables and change in loneliness was not consistent across all thresholds of the change in loneliness variable. We created binary variables representing these thresholds: increase in loneliness versus no change/decrease in loneliness and increase/no change in loneliness versus decrease in loneliness. Logistic regression models showed that having more symptoms of anxiety or living alone was related to higher odds of being in the increase in loneliness category relative to the no change/decrease in loneliness category but that these variables were not significantly related to being in the increase/no change category versus the decrease category.

Our aim was to assess changes in psychosocial factors and health behaviours caused by the experience of lockdown. However, it is also possible that the changes documented in this study reflect ageing effects, occurring between ages 82 and 84. To examine this possibility, we tested whether ageing effects might result in changes of a similar magnitude to those observed in the present study. We tested for changes in social support and physical activity occurring between two previous waves of the LBC1936 study (before the COVID−19 pandemic): Wave 4 (mean age 79) and Wave 5 (mean age 82; baseline in the main analysis). Wilcoxon signed rank tests indicated nonsignificant changes in social support *p* = 0.360 and physical activity *p* = 0.129 during this time. This comparison is shown in Appendix A.

## 5. Discussion

In this study we used repeated measures of physical activity, sleep quality, and psychosocial variables to examine longitudinally the potential impact of COVID−19 lockdown on a cohort of community dwelling 84-year olds. In exploratory analysis, we observed a small increase in the number of participants doing only minimal physical activity during lockdown and a small increase in perceived levels of social support. Changes in psychosocial and behavioural variables were modestly related to one another—those who experienced negative changes in one domain were slightly more likely to experience negative changes in other domains too. We identified individual differences related to more positive or negative change in the outcome variables during lockdown. Personality variables (higher levels of intellect, emotional stability, and extraversion) and higher cognitive ability were associated with more positive change during lockdown. By contrast, participants with a history of cardiovascular disease, more symptoms of anxiety, or living alone were likely to experience more negative changes.

Our observation of lower levels of physical activity during lockdown is in line with some other reports on adults [1] and children [2] that indicate a decrease in vigorous physical activity and an increase in sedentariness during this time. However, there is also evidence from research with adults and from consumer reports that participation in online exercise classes, home workouts, and time spent walking increased during lockdown [1,52]. Our finding of a decrease in physical activity could suggest that such online or home-based physical activities are less accessible to older people. This trend may have important implications for future cognitive and health trajectories of older adults. Physical activity and function are positively related to cognitive and brain health in older people [53,54,55] and protect against the risk of frailty [56].

Our findings also showed that compared with before lockdown, participants perceived themselves to have slightly higher levels of social support. This positive change could reflect an increase in online or phone contact with family and friends. According to a systematic review by Tajvar, Fletcher, and Grundy [57], there is evidence that social support has a moderate protective effect on mental health in older people. This association was also observed in the present study as more positive change in social support was correlated with more positive change in wellbeing. It is possible that an increase in social support may help to alleviate some of the negative consequences of lockdown on older people’s mental health. However, we note that the positive change observed in the present study contrasts with warnings that older people are at a high risk of social isolation and loneliness during lockdown [58]. It is likely that the characteristics of participants included in the analytical sample, including high occupational social class, cognitive function, and internet use reduced their risk of becoming isolated.

Our study also illustrates how individual differences influence how well older people cope with the experience of lockdown. In particular, the personality traits emotional stability, intellect, and extraversion were variously associated with more positive changes in wellbeing, neighbourhood cohesion, physical activity, or social support during lockdown. This finding corroborates some other reports. For instance, Kroencke et al. [59] found that the personality trait neuroticism (the opposite of emotional stability) was associated with higher levels of negative affect during the COVID-19 pandemic. Participants from the manual occupational class reported a greater increase in social support than participants from the professional occupational class. This result was unexpected as reports of low social support are more frequent among economically disadvantaged groups [60,61]. However, this association remains to be explored in the context of the COVID-19 pandemic. Reporting a history of cardiovascular disease was associated with more negative change in wellbeing and neighbourhood cohesion during lockdown. There is some evidence that cardiovascular disease is associated with an increased risk of death from COVID-19 [62]; the more negative experience of participants with cardiovascular disease may therefore result from higher levels of concern or stricter social distancing measures in this group. Finally, living alone and symptoms of anxiety are typically associated with higher levels of loneliness; in line with some other reports [24,63], our results suggest that participants with these characteristics are at an even greater risk of becoming lonely during the COVID-19 pandemic.

Overall, the LBC1936 sample experienced few negative changes during the first national lockdown in Scotland, suggesting that this demographic (community dwelling older people with above average levels of healthiness, cognitive ability, and social resources) might represent a relatively resilient group in the face of the pandemic. However, even in this sample, participants with certain characteristics (lower cognitive ability, more symptoms of anxiety, a history of CVD, and living alone) were more likely to experience negative changes during lockdown. Older people with these characteristics may especially benefit from additional support during the pandemic. Interventions to reduce loneliness, facilitate social connection, and create opportunities for physical activities could be beneficial. Efforts to support older people could also take personality differences into account, individuals with lower emotional stability, intellect, and extraversion may experience the challenges of lockdown more acutely than others.

We note several limitations of our study that should be taken into consideration. Firstly, owing to the novelty of the field and multiple significance tests carried out, this is an exploratory study; the results described indicate the relationships observed in the LBC1936 sample and identify hypotheses to be tested in further research. Many of the associations described in this study did not reach an FDR-corrected significance threshold. Therefore, our results might not generalise to the wider population of older people. Furthermore, we were only able to collect responses from participants with internet access and a certain level of technology proficiency. We found significantly higher levels of age 11 IQ, years of education, occupational class status, cognitive scores, and the personality traits emotional stability and intellect among those who took part in the online survey compared with those in the LBC1936 who did not. The LBC1936 sample is characterised by higher levels of healthiness, cognitive ability, and socioeconomic resources than the general population of older people in Scotland. Nevertheless, there is substantial variance in these variables in the LBC1936 sample. Studies using LBC1936 data have identified a range of genetic, health and fitness, psychosocial, and lifestyle variables associated with healthy ageing. These findings have been replicated in studies with other more diverse samples of older people [64]. Individuals from Black, Asian, and Minority Ethnic populations, are not represented in this sample; this limitation is particularly relevant given that these populations have been disproportionately affected by COVID-19 [65]. Although we were able to test for changes across a range of psychosocial and behavioural variables, additional outcomes of interest (assessed previously as part of the LBC1936 study) including diet quality and symptoms of anxiety or depression were not assessed as part of the LBC1936 COVID-19 survey and therefore could not be included as outcome variables in the study. Finally, because assessment occasions (before and during lockdown) were ~2 years apart, it is possible that some changes were caused by the effects of ageing rather than lockdown measures. However, results from subsidiary analysis, testing for changes in physical activity or social support between earlier waves of the LBC1936 study, suggest that age-related changes are typically smaller than those documented in the present study.

## 6. Conclusions

This study provides novel insight into the experience of older people during the COVID-19 pandemic. Our results illustrate some interdependence between psychosocial and health behaviour variables and their changes during lockdown. Individuals coping less well during this time may experience negative changes across multiple domains (e.g., mental wellbeing, loneliness, and memory problems). Our results also highlight that older people should not be viewed as a homogenous group in this context. Individual differences including cognitive ability, personality, living situation, and medical history predicted whether participants would experience more positive or negative changes during lockdown. In combination with findings from other studies with older people, this information could be used to identify and support those at highest risk in the case of further lockdowns.

## Figures and Tables

**Figure 1 ijerph-18-00210-f001:**
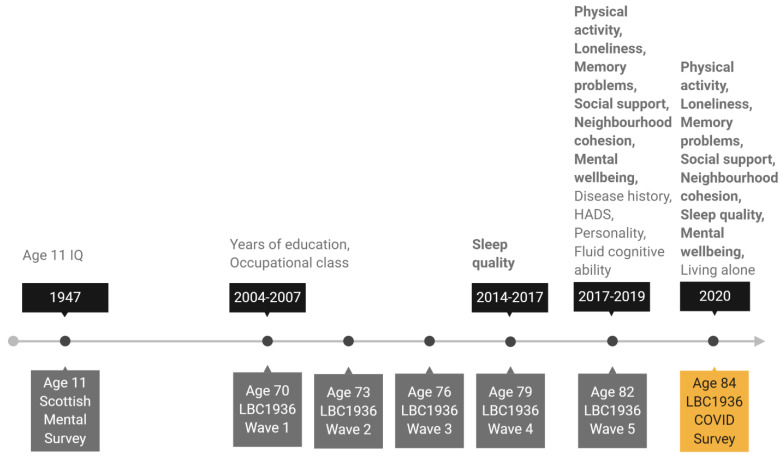
Timeline of the study showing when covariate and outcome variables were assessed. Note: Variables in bold are outcome variables assessed before and again during lockdown. All other variables are covariate variables. HADS: Hospital Anxiety and Depression Scale.

**Figure 2 ijerph-18-00210-f002:**
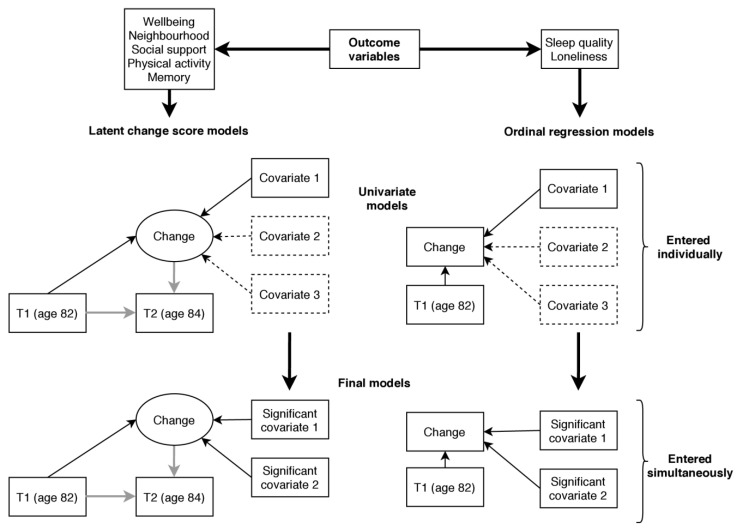
Illustration of the main analysis predicting change in the outcome variables. Note: Rectangles represent observed variables and ellipses latent variables. Single headed arrows represent regression paths. Grey arrows show regressions fixed at 1 (for the identification of the latent change score). Covariates 2 and 3 in the dashed boxes in the univariate models indicate that a list of covariate variables were examined in univariate analysis, each in a separate model (11 covariates were examined in total, listed in Section 3.3). Significant covariates 1 and 2 in the final model indicate that all significant variables identified in univariate analysis were included in the final model simultaneously.

**Figure 3 ijerph-18-00210-f003:**
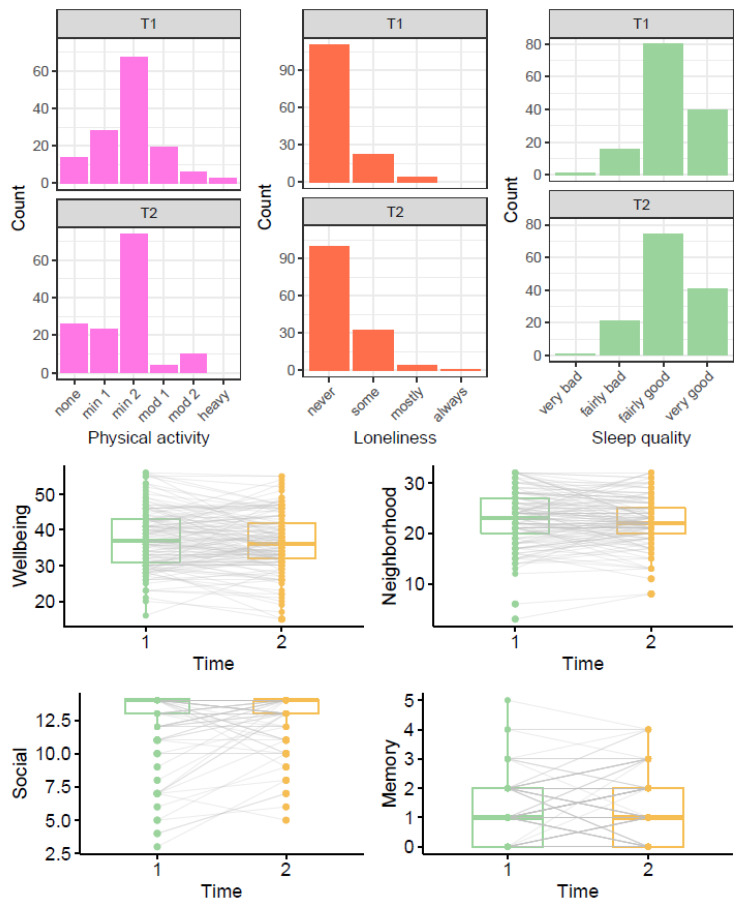
Comparison of outcome variables before (T1) and during lockdown (T2). Note: In the boxplots, grey lines connecting T1 and T2 points show trajectories for each individual. For physical activity, none = only household chores, min1 = outdoor activities 1–2x per week, min2 = outdoor activities > 2x per week, mod1 = moderate exercise 1–2x per week, mod2 = moderate exercise > 2x per week, heavy = keep-fit/heavy exercise severalx per week.

**Table 1 ijerph-18-00210-t001:** Comparison of outcome variables before (T1) and during lockdown (T2).

	T1	T2	*p*
Sleep quality			0.597 ^1^
-very bad	1 (0.7%)	1 (0.7%)	
-fairly bad	16 (11.7%)	21 (15.3%)	
-fairly good	80 (58.4%)	74 (54.0%)	
-very good	40 (29.2%)	41 (29.9%)	
**Physical activity**			0.012 ^1^
-only household chores	14 (10.2%)	26 (19.0%)	
-outdoor activities 1–2x per week	28 (20.4%)	23 (16.8%)	
-outdoor activities > 2x per week	67 (48.9%)	74 (54.0%)	
-moderate exercise 1–2x per week	19 (13.9%)	4 (2.9%)	
-moderate exercise > 2x per week	6 (4.4%)	10 (7.3%)	
-keep-fit/heavy exercise several times per week	3 (2.2%)	0 (0.0%)	
Loneliness			0.059 ^1^
-none/almost none of the time	111 (81.0%)	100 (73.0%)	
-some of the time	22 (16.1%)	32 (23.4%)	
-most of the time	4 (2.9%)	4 (2.9%)	
-all or almost all the time	0 (0.0%)	1 (0.7%)	
Memory problems			0.192 ^1^
-Mean (SD)	1.212 (1.046)	1.321 (1.156)	
Wellbeing			0.076 ^2^
-Mean (SD)	37.453 (8.369)	36.453 (8.230)	
Social support			0.032 ^1^
-Mean (SD)	12.759 (2.490)	13.095 (1.802)	
Neighborhood cohesion			0.108 ^2^
-Mean (SD)	22.920 (5.377)	22.372 (4.756)	

Note: ^1^ Wilcoxon signed rank test, ^2^ Paired sample *t*-test.

**Table 2 ijerph-18-00210-t002:** Correlation between changes in outcome variables.

	Sleep	Physical	Loneliness	Memory	Wellbeing	Social Support
Sleep						
Physical activity	−0.161					
Loneliness	−0.102	−0.095				
Memory	0.035	−0.023	0.100			
Wellbeing	0.030	−0.035	−0.220 **	−0.223 **		
Social support	−0.099	−0.064	−0.096	−0.169 *	0.174 *	
Neighbourhood	−0.018	−0.119	−0.006	−0.038	0.138	0.135

Note: Correlations are Spearman’s rho. * *p* < 0.05, ** *p* < 0.01.

**Table 3 ijerph-18-00210-t003:** Results from final models predicting change in the outcome variables.

Change Variable	Covariate	Est	95% CI	*p*	FDR *p*
Wellbeing ^a^	**Emotional stability ***	0.221	0.094, 0.348	0.001	0.003
	Living with others	0.252	−0.031, 0.536	0.081	0.081
	**History of CVD ***	−0.376	−0.634, −0.118	0.004	0.006
Neighbourhood ^a^	**Intellect**	0.147	0.022,0.272	0.021	0.025
	**History of CVD**	−0.310	−0.580, −0.039	0.025	0.025
Social support ^a^	**Intellect**	0.118	0.003, 0.232	0.044	0.088
	Occupational class				
	managerial	0.195	−0.063, 0.453	0.138	0.184
	skilled non-manual	−0.052	−0.461, 0.357	0.804	0.804
	**manual**	0.575	0.192, 0.959	0.003	0.012
Physical activity ^a^	Intellect	0.035	−0.102, 0.173	0.615	0.718
	**Extraversion**	0.162	0.035, 0.289	0.012	0.084
	**Age 11 IQ**	0.143	0.017, 0.269	0.026	0.091
	Fluid g	0.137	−0.014, 0.287	0.075	0.175
	Occupational class				
	managerial	−0.018	−0.289, 0.254	0.899	0.899
	skilled non−manual	0.235	−0.232, 0.702	0.324	0.454
	manual	−0.302	−0.682, 0.077	0.118	0.206
Memory ^a^	**Depression**	−0.159	−0.316, −0.002	0.047	−
Loneliness ^b^	**Anxiety**	1.222	1.049, 1.424	0.010	0.010
	**Living with others ***	0.149	0.061, 0.364	<0.001	0.002

Note: Variables under each outcome heading are entered simultaneously. Occupational class is dummy coded with professional as the reference category. Estimates are standardized *β*; for binary variables, estimates represent a change in the dependent variable in standard deviation units when the binary covariate changes from zero to one. For the Hochberg’s False Discovery Rate (FDR) *p*, *p*-values from each final model were grouped and an FDR correction was applied to each group of *p*-values separately. ^a^ Estimates from latent change score models additionally adjusting for outcome variable at T1; ^b^ Estimates are odds ratios (ORs) from an ordinal logistic regression model predicting change in loneliness (increase, no change, or decrease) and adjusting for loneliness at T1 (none of the time vs. some of the time or more). Bold typeface denotes *p* < 0.05. * Variables that survive FDR correction in the univariate and final models

## Data Availability

LBC1936 data are available upon request. To request the data, readers should contact the LBC Director S.R.C.: simon.cox@ed.ac.uk. Details regarding the analytic methods including R and Mplus scripts can be requested from the corresponding author: J.A.O.: judith.okely@ed.ac.uk. The data are not publicly available due to ethical restrictions.

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
