# Peer review of "Change in Physical Activity, Sleep Quality, and Psychosocial Variables during COVID-19 Lockdown: Evidence from the Lothian Birth Cohort 1936"

_ijerph, 2020, doi:10.3390/ijerph18010210_

Round 1

Reviewer 1 Report

I find the present work of high quality and current relevance. It explores how the lockdown and protective measures imposed in response to the pandemic can affect the health and wellbeing of older people (mean age 84). Available evidence suggests that, in times like the ones we are living, this group is at greater risk of being more severely affected, both physically and psychologically.

The main strength of this research is its longitudinal approach. Unlike other works on this topic (which are mainly cross-sectional), authors provide data which were collected at two different moments, before and during COVID-19 lockdown.

As to the manuscript itself, it is well structured and easy to follow. The introduction briefly summarizes the main knowledge available on the topic and demonstrates the novelty and relevance of the conducted research. The methodology (including the analyses carried out) is clear and properly justified. Finally, the results and discussion sections present, describe and interpret the main findings of the research conducted.

Since the Introduction section was concluded indicating that the current research was expected to provide information regarding "how to protect those who are more vulnerable" during pandemic time, I  think the authors could have discussed in greater depth the meaning of the findings and how these findings could be applied. However, as I indicated before, it is an innovative work, of good methodological quality and great public interest. Therefore, I highly recommend its publication.

Reviewer 2 Report

This is an interesting study that deserves to be published, after some major review:

  • Abstract: clarify the method and results. Many variables are mentioned and the results are confusing.
  • Introduction: a more detailed literature review is needed, and, mainly, why is important to study these variables in a sample of elderly people.
  • method:

. Participants: a more detailed sample characterization is needed. if T1 was measured at mean age 82 years-old, how authors explain the sentence “this also includes participants who did not participate in the most recent wave of the LBC1936 study at mean age 82”. This is related to variable sleep quality?

. Health behaviour variables should be replaced by Physical activity and Sleep quality (Health behaviour variables is too vast)

. why did authors choose for covariates sex, age (participants have the same age, right?)? symptoms of depression and anxiety should be considered a dependent variable

. the outcome variables should be well-defined. Why not include depression and anxiety as outcome variables, if authors aim to analyse behavioural and psychosocial consequences of lockdown??

. Figure 1 (Illustration of the main analysis predicting change in the outcome variables) is less informative for covariates 1, 2, and 3, and also for significant covariate 1 and 2… authors should specify inside the figure

  • Results:

. Figure 2 - Comparison of outcome variables before (T1) and during lockdown (T2) – ordinal variable Physical activity should be revised in order to perform Wilcoxon signed-rank test properly

  • . the analysis referring to the lower levels of physical activity during lockdown should be repeated, maybe using a different test (chi-square), in order to test the accuracy of the conclusion reached by the authors

. Authors should present and discuss the effect size of the differences and of the correlations. For example, r = -0.169 has only an effect size of R2 = 2.85%.

. authors should be more clear in some sentences (for example: “symptoms of anxiety and living alone were only related to an increase in loneliness relative to no change or a decrease in loneliness.” )

- discussion: it is important to analyse the external validity of this study since elderly people of Lothian Birth Cohort could not be representative of other elderly people. It is important to revise previous studies with LBC1936 participants in order to see if their outcomes are similar to other elders  outside LBC1936

Reviewer 3 Report

This research has an original objective and a meaningful content. I congratulate the authors for the work done. I am grateful with the editors for the possibility of revising this manuscript. Although the quality of the manuscript is high, I would like to make some contributions that I hope will increase it and improve readers' understanding.

Introduction

The introduction is clear and well worked.

Materials and Methods, appropriate.

2.2LBC1936 study assessments a flow chart would greatly improve the explanation

Study design

The study design is appropriate and well described.

Results:the first three lines do not contribute anything, it is already described above, they are already described in point 3.4
